# A Risk Profile for Disordered Eating Behaviors in Adolescents with Type 1 Diabetes: A Latent Class Analysis Study

**DOI:** 10.3390/nu15071721

**Published:** 2023-03-31

**Authors:** Giada Boccolini, Monica Marino, Valentina Tiberi, Antonio Iannilli, Giulia Landi, Silvana Grandi, Eliana Tossani, Valentino Cherubini

**Affiliations:** 1Department of Women’s and Children’s Health, G. Salesi Hospital, 60123 Ancona, Italy; 2Laboratory of Psychosomatics and Clinimetrics, Department of Psychology, University of Bologna, 47521 Cesena, Italy

**Keywords:** disordered eating behaviors, type 1 diabetes, nutritional habits, adolescents, latent class analysis

## Abstract

(1) Background: This multi-center study aimed to identify a risk profile for disordered eating behaviors (DEBs) in youth with type 1 diabetes (T1D) based on their dietary intake, lipid profile, body mass index (BMI-SDS), and glycometabolic control. (2) Methods: Adolescents aged 11 to 18 years from five centers across Italy were recruited. Lipid profile, HbA1c, BMI-SDS, and dietary intake data were collected. The risk for developing DEBs was assessed via the Diabetes Eating Problems Survey-R (DEPS-R) questionnaire. A latent class analysis (LCA) was performed using a person-centered approach. (3) Results: Overall, 148 participants aged 11–18 (12.1, ±3.34), 52% males with a mean diabetes duration of 7.2 (±3.4), were enrolled. Based on the results of the DEBS-R score, LCA allowed us to highlight two different classes of patients which were defined as “at-risk” and “not at-risk” for DEB. The risk profile for developing DEBs is characterized by higher BMI—SDS (23.9 vs. 18.6), higher HbA1c (7.9 vs. 7.1%), higher LDL cholesterol (99.9 vs. 88.8 mg/dL), lower HDL cholesterol (57.9 vs. 61.3 mg/dL), higher proteins (18.2 vs. 16.1%), and lower carbohydrates (43.9 vs. 45.3%). Adolescents included in the “at-risk” class were significantly older (*p* = 0.000), and their parents’ SES was significantly lower (*p* = 0.041). (4) Conclusions: This study allowed us to characterize a risk profile for DEBs based on dietary behavior and clinical parameters. Early identification of the risk for DEBs allows timely intervention and prevention of behavior disorders.

## 1. Introduction

Disordered eating behaviors (DEBs), including diagnosable eating disorders (Eds), in the context of type 1 diabetes (T1D), are prevalent and can directly interfere with optimal diabetes management [1,2,3]. A recent Italian cross-sectional study of 690 adolescents with T1D estimated a DEB prevalence of 28.1% (21% boys, 35% girls) [4]. Similar results have been reported for other adolescent populations with T1D across the world [5,6,7,8,9]. A population-based study from national registers in Sweden and Demark investigated the within-individual association between T1D and eating disorders, highlighting a risk of 2% of developing a subsequent eating disorder in this population [10]. DEBs cannot be categorized as proper diseases but they are considered as mild symptoms that can evolve into EDs, and include behaviors such as dieting for weight loss, binge eating, self-induced vomiting, excessive exercise, and laxative or diuretic use. Young people with T1D reported another unique way of controlling body weight, the voluntary reduction or omission of insulin therapy [11]. These maladaptive behaviors are associated with worse glycemic control in children and adolescents with T1D [12]. Indeed, most of the DEBs reported in this population, such as skipping meals or binge eating, were associated with higher HbA1c values. Prolonged hyperglycemia and elevated HbA1c caused by DEBs can accelerate long-term diabetes-related complication onset, including retinopathy, nephropathy, and neuropathy [13]. In addition, the risk of developing DKA must be considered, as insulin omission is a common strategy to induce hyperglycemia and weight loss in individuals with T1D and DEBs. Another recent Italian study identified a clinical profile for adolescents with T1D positive to DEB screening characterized by being overweight, having little time spent performing physical activity, a low socioeconomic status, poor metabolic control, and skipping insulin injections [14]. In the study, overweight youth were six times more likely to report DEBs. Other studies in the adult population of T1D highlighted how overweight and obesity increased the prevalence of subclinical eating disorders from 3.6% to 15–20% [15]. Children and adolescents with T1D have issues with food and diet. Overweight and obese youth and young adults with diabetes are commonly counseled to reduce their weight. Binge eating is frequent in this population, and it is associated with a dietary pattern characterized by the intake of hyperpalatable foods rich in sugars and fats [16,17]. Nevertheless, it has been largely shown that adolescents with T1D generally fail to meet recommended nutrient intakes as outlined in the International Society for Pediatric and Adolescent Diabetes (ISPAD) dietary guidelines for T1D [18,19,20]. Cross-sectional studies report that the overall intake of total and saturated fats is high, while the intake of fruits, vegetables, and grains is low [21,22]. Although many descriptive analyses have been performed to define the dietary pattern in this population, there has been a growing interest in the application of latent class analysis (LCA) in the psychological clinical and nutritional fields, as it appears to be an efficient instrument for exploratory investigations and the identification of risk profiles. LCA is a statistical analysis adopting a person-centered approach that aims to detect similarities in given characteristics—clinical, psychological, socioeconomic, etc.—within a certain sample of individuals. The LCA methodology takes into account differences between individuals while unveiling latent associations among them through the creation of classes. Each resulting class is composed of individuals that share similar features according to the characteristics of interest. To the best of our knowledge, no studies highlight nutritional habits and dietary intake in the context of DEBs in T1D relying on LCA. The main aim of the present study was to evaluate the association between dietary intake, lipid profile, BMI-SDS, glycometabolic control, and DEBs in adolescents with T1D in order to identify a risk profile for this population.

## 2. Materials and Methods

This multicenter study analyzed data collected from five Italian pediatric diabetes centers (Torino, Verona, Ancona, Roma, and Napoli) that were willing to collect data on CGM glucose metrics and ask included patients to complete a food diary and a screening questionnaire for disturbed eating behaviors. Inclusion criteria for participants were a diagnosis of T1D for over six months, being aged from 2 to 17 years, using Dexcom G6 CGM System (Dexcom, Inc., San Diego, CA, USA) for over six months, having an active connection using Clarity^®^ software (Dexcom international Switzerland, Horw, Switzerland), an HbA1c < 10% (86 mmol/L) during the three months prior to recruitment, and parents available to collect and record nutritional information for three days. Exclusion criteria were the usage of a type of blood glucose monitoring other than the Dexcom G6 CGM, unwillingness or inability of the parents to fill a three-day food diary, previous diagnosis of celiac disease, and HbA1c > 10% at study recruitment. Children aged 11–17 were also asked to fill out the Diabetes Eating Problems Survey-R (DEPS-R) questionnaire to assess the risk of DEBs in this study population. This study was approved by the Independent Ethics Committees of all five participating centers.

### 2.1. Study Procedures

The present study analyzed data collected before the start of the SARS-CoV-2 pandemic in Italy, from January 2019–January 2020. All patients who met the eligibility criteria and their families were informed about the possibility to participate in the study during a scheduled follow-up visit at the pediatric diabetes center. The personal, clinical, and laboratory data regularly recorded during the medical visit at the diabetes center were collected for the study purposes, after obtaining a signed informed consent from the study participants and their parents. Further information about the level of education and occupation of the parents was collected to define their socioeconomic status (SES). At the same medical visit, study participants aged 11–17 were asked to fill out the DEPS-R questionnaire for DEB screening. In addition, a trained dietitian in each center provided the families enrolled in the study with a kitchen scale and a food diary for the dietary intake and nutritional habits assessment. Each family was also provided with specific training regarding how to fill out the food diary.

### 2.2. DEPS-R Questionnaire

The DEPS-R is a T1D-specific measure for DEBs that includes weight loss, insulin dose omissions, dietary restrictions, and induced vomiting [2]. The instrument is a self-report questionnaire including 16 items. Questionnaire items are rated on a 6-point Likert scale ranging from 0 (never) to 5 (always) to quantify the frequency of DEBs. DEPS-R scores of 20 or higher indicate higher risks for developing DEBs [2,7]. In addition, the DEPS-R questionnaire can assess the prevalence of DEBs through T1D-specific compensatory behaviors regarding the maintenance of high blood glucose to lose weight [7]. The DEPS-R has demonstrated satisfactory psychometric properties [6,7] in Italian samples [4,14] with good internal consistency (Cronbach’s alpha > 0.8), test–retest reliability (inter-class correlation coefficient > 0.9), and convergent and criterion validity. In the present study, the Italian version of the DEPS-R [14] was used to assess DEBs in adolescents 11–17 years of age. The observed Cronbach’s alpha in this sample was 0.586.

### 2.3. Dietary Intake Assessment

The dietary intake was assessed using the DONALD study’s three-day weighed dietary record [23]. A trained dietitian advised study participants to maintain their current daily eating habits during the three days of recording. Parents of patients included in the study were carefully instructed on how to collect data and how to evaluate foods and record data using the kitchen scale (Soehnle Digital, Soehnle Professional, Backnang, Germany) provided by the pediatric diabetes center. They were asked to record food diaries for three days (Sunday, Monday, and Tuesday) starting the weekend following recruitment. The information requested in the food diary was the type and brand of food, the quantity of food consumed, estimated in grams or semi-quantitative measurements, the time and place of consumption, and any recipes used. For commercial food items, the packages or the food labels were collected, and the product information was added to the dietary record using Winfood^®^ software version 3.0 (Medimatica, Teramo, Italy).

### 2.4. Family SocioEconomic Status

Family characteristics such as parents’ age, education, occupation, and history of diabetes were collected. Parents’ education was categorized into seven levels: less than 7th grade, junior high/middle school (9th grade), partial high school (10th or 11th grade), high school graduate, partial college (at least one year), college education, and graduate degree. The classification of professions of the Italian Institute of Statistics (ISTAT) [24] was used to classify parents’ occupations according to nine major groups; occupational categories were considered in two skill levels: high (Manager, Legislators, Chief Executives Officials, Technicians and Associate Professionals, Science, Engineering, Health, Teaching, Business and Administration, Information and Communications Technology, Legal, Social, Cultural Professionals, and Armed Forces Officers) and low (Elementary Occupations, Clerical Support Workers, Services and Sales Workers, Skilled Agricultural, Forestry and Fishery Workers, Craft and Related Trades Workers, Armed Forces Occupations, and Other Ranks). These data were summarized and scored using the Barratt Simplified Measure of Social Status (BSMSS) [25]. Parental familiarity with T1D and type 2 diabetes (T2D) was also investigated.

### 2.5. Nutritional Habits and Glycemic Control Variables

All centers used the same analytical laboratory methods for lipid profile data collection. Total cholesterol, HDL cholesterol, and triglycerides were measured in stored plasma samples by an enzymatic method using a Beckman Coulter Olympus AU 480 (Beckman Coulter, Brea, CA, USA), and LDL was calculated using the Friedewald equation, which includes total cholesterol, HDL cholesterol, and triglycerides. BMI was calculated from registered height and weight as weight/squared height (kg/m^2^) and converted to BMI-SDS (standard deviation score) using WHO growth curves [26]. A dietary assessment was performed by the same dietitian (MM) for all centers using the three-day weighted food diary. Dietary intake was assessed by calculating the total daily kilocalorie intake and the percentages of sugars, total carbohydrates (CHO), saturated fatty acids (SFAs), monounsaturated fatty acids (MUFAs), polyunsaturated fatty acids (PUFAs), and protein intake in diets. HbA1c was measured with the DCA Vantage^®^ Analyzer.

### 2.6. Other Variables

Further clinical and demographic characteristics of the youth included date of birth and date of diabetes diagnosis, gender, type of insulin therapy (multiple daily injections—MDI, Continuous Subcutaneous Insulin Infusion—CSII), average total daily insulin dose, and the use of a carbohydrate counting (CC) method to calculate daily insulin dose. Glucometrics assessed by CGM data (time in range (TIR), percentage of time < 54 mg/dL, 54–70 mg/dL 180–250 mg/dL, and >250 mg/dL, coefficient of variation, and the percentage of time that CGM was active) were collected during the 15 days following the recruitment visit. Physical exercise was assessed by the self-reported number of weekly hours spent in activity by the study participants.

### 2.7. Statistical Analysis

LCA is a statistical method that relies on a person-centered approach in order to evaluate the degree to which different characteristics co-occur within individuals and whether subgroups (or classes) are detectable [27,28]. As the aim of this study was to identify a risk profile for the potential development of DEBs in adolescents with T1D, we expect classes of individuals to display hidden patterns of different risk profiles based on nutritional habits, lipidic profiles, and DEPS-R scores. Variables included in the models were BMI-SDS, HbA1c, macronutrient values (proteins, carbohydrates, sugars, saturated fats, polyunsaturated fats, and monounsaturated fats), lipid profile (LDL and HDL, and triglycerides), and DEPS-r score. LCAs were conducted with Mplus 8.329. After the identification of the clusters, ANOVAs and chi-square analyses were run in SPSS25 (IBM Co., Ltd., Chicago, IL, USA) in order to investigate differences between individuals belonging to potentially different profiles. Data are summarized as mean and standard deviation (SD). A level of probability of 0.05 was used to assess the statistical significance.

## 3. Results

### 3.1. Descriptive Analyses

A total of 197 children with T1D were enrolled in this study. As DEPS-R is only validated for adolescents 11–17 years old, participants aged < 11 did not fill out the questionnaire; therefore, they were excluded. Moreover, participants whose clinical data were unavailable were excluded as well. Finally, 148 adolescents with T1D were included in these analyses (Figure 1).

As depicted in Table 1, 98% of the sample was of Italian nationality, while 2% was African American. Participants’ mean age was 12.1 (3.3), with a minimum age of 3.3 and a maximum of 17.8. With regards to gender, 48% were female and 52% were male. The mean duration of diabetes was 7.2 years (3.4), with a minimum duration of 1.0 years and a maximum of 16.3 years. As for the modality of insulin therapy adopted, 47.9% of patients used CSII, while 52.1% used MDI. The majority of our sample (23.1%) reported engaging in 2 h of physical exercise per week, followed by 17.7% who engaged in 3 h of physical exercise per week, and 15% who reported no exercise at all during the week. Of this sample, 46.3% reported utilizing CC, while 53.7% did not. The medium insulin dose reported was 37.8 UI/die (19.2). CGM data for adolescents included in these analyses revealed a TIR of 57.9%. Other glucose metrics collected are reported in Table 1. With regard to the DEPS-R scores, although the sample would predominantly fall below the cut-off of 20 points, the analyses identified 24 patients (16.2% of the sample) with a score above the cut-off, who are thereby identified as potentially at risk for developing DEBs. Similar results are exhibited in BMI-SDS scores and HbA1c values, as 84.5% of the population displayed a BMI-SDS in line with the values expected for their age, height, and weight, while 15.5% were overweight. Data emerging from three-day food diaries are reported in Table 2. Adolescents included in the present study consumed a mean percentage of macronutrients as follows: 45.05% CHO, 10.97% sugars, 16.78% proteins, 9.25% SFA, 16.44% MUFA, and 9.91% PUFA. Education and occupational status were investigated for both parents and are displayed in Table 3. The majority of both mothers and fathers reported having a high school diploma as their highest educational achievement, 59.5% and 50.3%, respectively; 29% of mothers and 23.8% of fathers had a university degree, whereas 11.5% and 25.9%, respectively, had a middle school diploma. With regards to occupational status, the sample for this study was heterogeneously distributed, although the highest percentage of parents (13.5%) reported working as skilled constructors, artisans, technicians, sales counters, or general office clerks. We also analyzed family income as a dichotomous variable to have an additional indicator of SES. According to the ISTAT investigations [24], the average yearly family income in Italy is EUR 29,300. We asked participants to indicate whether they would identify their family income to be placed above or below this threshold. A total of 62.2% of the study population was above the threshold, while 37.8% was below. Familiarity with both T1D and T2D diabetes was also explored. For the most part, parents did not display any familiarity with either T1D or T2D, 93.2% and 88.5%, respectively. Nonetheless, a few cases of familiarity were evinced, with 6.8% of parents reporting T1D and 11.5% reporting T2D.

### 3.2. Latent Class Analysis

As previous studies suggested, an estimation of the latent class was performed using robust maximum likelihood estimation, and we selected the final model considering that the sample size was adjusted with Bayesian information criterion (SSA-BIC), the adjusted Lo–Mendell–Rubin Likelihood Ratio Test (LMR-LRT), entropy values, clinical significance, and satisfactory size of each class. Lower SSA-BIC, a significant value of LMRT-LRT (*p* < 0.05), and entropy values higher than 0.80 were indicative of the most accurate model [29,30]. A two-class model was selected, as the LMR-LRT index was not significant, suggesting that adding an additional class would have meant losing statistical significance for the model. Through observation and clinical evaluation of the two classes, the first one was labeled as “not at—risk” and the second one as “at—risk”. LCA results are summarized in Table 4. Of the total sample, 114 participants were included in the “not at—risk” class. Adolescents in this cluster appeared to have lower mean values of BMI-SDS, HbA1c, LDL, proteins, monounsaturated fats, and DEPS -R. In addition, they displayed higher mean values of HDL, saturated fats, polyunsaturated fats, carbohydrates, and sugar. On the other hand, 34 participants were included in the “at—risk” class. Adolescents in this cluster appeared to have higher mean values of BMI-SDS, HbA1c, LDL, proteins, monounsaturated fats, and DEPS -R; and they appeared to have lower mean values of HLD, saturated fats, polyunsaturated fats, carbohydrates, and sugar (see Figure 2).

### 3.3. Differences in Youth’s Age, Parental SES, and Diabetes-Related Variables between Classes

Following LCA and the identification of the classes, further ANOVA analyses were performed to determine whether significant differences would emerge in participants’ age, parental SES, and diabetes-related variables. We found statistically significant differences between the mean age of children in the “not at—risk” class and children in the “at—risk” class (*p* ≤ 0.000). The mean age in the “not at—risk” class was 11.37 (3.30), while in the “at—risk” class, participants were significantly older, with a mean age of 14.65 (1.97). The “not at—risk” class adolescents demonstrated to have significantly higher mean scores for parental SES with respect to “at—risk” class adolescents (*p* ≤ 0.05), 37.53 (14.27) and 32.09 (10.34), respectively. Data around episodes of severe hypoglycemia and DKA in the last 12 months have shown no significant differences between the “not at—risk” and “at—risk” classes, potentially due to the low incidence of these occurring during the study period. In fact, 97.3% of our sample never had any episodes of severe hypoglycemia in the year of data collection. Similarly, only three (2%) participants were hospitalized for an episode of DKA, showing that the great majority of the study population (98.0%) never had an episode of DKA in the year of data collection. As for the analysis around the number of insulin injections and the number of omitted insulin doses, no significant differences were detected, while daily insulin dose was significantly different between the two clusters. In particular, adolescents in the “at—risk” cluster demonstrated to have a mean value of daily insulin dose significantly higher (*p* ≤ 0.05) than those in the “not at—risk” cluster, 47.57 (20.53) and 34.82 (17.89), respectively. There was not a statistically significant difference between the clusters in terms of weekly hours of physical exercise. When looking at the CGM data for the percentage of time spent below, in, and above the range, a statistical difference was displayed for the time spent severely below the range (<54 mg/dL), below the range (54–70 mg/dL), and severely above the range (>250 mg/dL). As expected, adolescents belonging to the “at—risk” cluster spent 0.31% of the time severely below the range, while adolescents in the “not at—risk” class spent more time in this range, 0.69% (*p* ≤ 0.05). In line with this result, a more evident difference emerged on the TBR. Adolescents in the “not at—risk” class spent significantly more time below the range (*p* ≤ 0.05) than the “at—risk” class adolescents, 3.07% and 1.93%, respectively. As predicted, the difference in percentages for TAR showed how adolescents belonging to the cluster at risk for DEBs spent significantly more time with severe hyperglycemia (*p* ≤ 0.05), 18.11%, as opposed to the 13.27% for the “not at—risk” class. These results are reported in Table 5.

### 3.4. Differences in Youth’s Gender, Carbohydrate Counting, and Modality of Insulin Therapy between Classes

Chi-square analyses were performed in order to evaluate differences between the two classes in the following categorical variables: gender (M vs. F), CC (Yes vs. No), and type of insulin therapy adopted (MDI vs. CSII). No statistically significant differences emerged between the classes with regard to these variables. Nonetheless, we observed changes in the numerosity itself between the clusters when focusing on these three characteristics. We noticed an equal distribution of males and females in both the “at—risk” and “not at—risk” groups, 55 females in the “not at—risk” class and 16 in the “at—risk” class versus 59 females and 18 males, respectively. We noticed that in the “at—risk” cluster, only 12 individuals reported relying on CC, while 22 did not. No significant differences were reported for insulin therapy modality in the two classes. These results are depicted in Table 6.

## 4. Discussion

The LCA analysis highlighted two classes that identify patients as “at—risk” and “not at—risk” on the basis of the mean values of DEPS-R, BMI-SDS, HbA1c, and dietary intake. The risk profile for developing DEBs is characterized by adolescents displaying higher BMI-SDS values, as well as higher HbA1c levels. Higher levels of LDL cholesterol and lower levels of HDL cholesterol in the “at-risk class” were reported. Regarding dietary intake, higher mean levels of proteins and MUFA indicated a potentially detrimental effect for developing DEBs as well. Moreover, adolescents who were “at—risk” also seemed to have a poorer diet poorer SFA, PUFA, CHO, and sugars, than those allocated in the “not at—risk” class. In addition, parents of adolescents in the “at—risk” class displayed a significantly lower SES than those parents of adolescents in the “not at—risk” class. The simultaneous presence of all these characteristics strongly suggests an “at—risk” profile for DEBs (see Figure 3). No statistical significance was detected in chi-square analyses performed to investigate differences between the two classes in terms of gender, therapy administration modality, carbohydrate counting, and familiarity of T1D and T2D.

In the present study, 16.5% of the participants enrolled were positively screened for DEBs. This result is comparable with data collected by cross-sectional studies in children and adolescents with T1D from all over the world (17.4% for Egyptian [5], 25% for Turkish [6], 13.8% for Norwegian [7], 39.3% for Chinese [8], 30.1% for Greek [9], and 28.1% for Italian populations [4]). Results from this study are in line with what the available literature specifies around adolescence and T1D [31], meaning that this age is the most difficult in terms of psychosocial adjustment to the illness as well as correct adherence to therapy. In fact, adolescents belonging to the “at—risk” class were significantly older than adolescents in the “not at—risk” class, with the former group displaying a mean age of around 14 years and the latter of 11. This result suggests that approaching adolescence represents not only a risk factor for adherence to therapy itself, but also for the potential development of DEBs [11,12,17,32]. Moreover, it is essential to notice that youth that are part of the “at—risk” class appeared to have had T1D for longer, as the mean illness duration was about 9 years, compared to 7 years for adolescents belonging to the “not at—risk” class. In this study, diabetes duration and adolescence appear to function as concurrent factors for adverse outcomes around DEBs. Adolescents with early onset T1D develop a deep understanding of the disease, as they have been struggling with diabetes for longer; therefore, the experience gained allows for greater independence in managing T1D. In this age, desires for both independence from the family and inclusion in the peer network reach their peak, and the concerns around body image as well efforts around acceptance of the T1D increase the difficulties in dealing with everyday life in this population [33]. Moreover, adolescents living with T1D are not seldomly discriminated against by their peers because of their condition, and it would not be unlikely for them to develop feelings of insecurity and low self-esteem. Consistently with these considerations, LCA analyses in the present study highlighted a significant difference in BMI-SDS levels, as the “at—risk” class displayed a mean BMI-SDS of 23.858 (0.950) vs. 18.579 (0.242). The association between higher BMI-SDS scores and the presence of DEBs has already been acknowledged in adolescents with T1D [4,14,34,35,36], considering weight as a predictive factor for the onset of DEBs in this population [32]. Indeed, the hypothesis of the association between dissatisfaction with body weight and omission of insulin dose has been already formulated in previous studies [37]. In addition, body weight dissatisfaction is correlated with loss-of-control eating behaviors in adolescents, suggesting problems with self-regulation in this population [38]. In youth with type 1 diabetes, there is a strong relationship between food intake, insulin therapy, and metabolic control that requires self-regulation. Moreover, self-regulation is often imposed by carbohydrate counting usage, increasing vulnerability to the development of DEBs [15]. The consequence of dietary dysregulation is the worsening of metabolic control in this population because of the adoption of unhealthy strategies (e.g., skipping meals, purging behaviors), that often involve insulin therapy (e.g., omitting insulin doses) [13]. Indeed, in the present study, the “at—risk” class reported higher levels of HbA1c, 7.867% (0.274) vs. 7.083% (0.071). Moreover, when looking at the CGM data for the percentage of time spent below, in, and above the range, a statistical difference was displayed for the time spent severely below the range (<54 mg/dL), below the range (54–70 mg/dL), and severely above the range (>250 mg/dL). As predicted, the difference in percentages for TAR showed how adolescents belonging to the at-risk cluster for DEBs spent significantly more time in severe hyperglycemia (*p* ≤ 0.05), 18.11% vs. 13.27% for the “not at—risk” class, bringing attention to diet and therapy dysregulation. As expected, adolescents belonging to the ‘“at—risk” cluster spent 0.31% severely below the range time, while adolescents in the “not at—risk” class spent more time in this range, 0.69% (*p* ≤ 0.05). In line with this result, a more evident difference emerged on the TBR, as participants in the “not at—risk” class spent significantly more time below the range (*p* ≤ 0.05) than “at—risk” participants, 3.07% and 1.93%, respectively. Impairment of metabolic control is also reflected in lipid profile status. In the present study, LCA analyses also highlighted differences in lipid profile, as the “at—risk” class reported higher values of LDL-cholesterol, 99.848 (5.382) vs. 88.808 mg/dL (1.952), and lower values of HDL-cholesterol, 57.845 (2.378) vs. 61.280 (1.221). The association between dyslipidemia and EDs has been previously reported in the literature [39]. In addition, a recent population-based Italian study reported higher levels of total cholesterol in children and adolescents positive for DEPS-R screening for DEBs [14]. However, there is a paucity of studies that evaluate lipid profile parameters as risk factors for developing DEBs, and this association needs to be confirmed by future research. Unexpectedly, in the present study, the included adolescents reported adequate dietary intake according to the ISPAD guidelines for nutrition in children with T1D [20,22,40,41], except for the total amount of lipids. Participants included in the “at—risk” class appeared to have a higher medium intake of proteins (18.157%) and MUFA (17.860%), and a lower intake of SFA (8.898%), PUFA (8.462%), CHO (43.993%), and sugars (10.605%). However, the percentages of macronutrients reported are still within the ISPAD recommendation range [42]. In the literature, the relationship between dietary intake and EDs in children and adolescents with T1D has already been investigated, underlying a significantly worse diet quality in children and adolescents with EDs [43]. Tse et al. used the HEI-2005 scale to describe diet quality based on food frequency in a three-day weighted food diary and Healthful Eating Attitudes Scale to define the nature of eating behaviors [43]. In the present study, the investigation was focused on nutritional habits (including dietary intake, BMI-SDS, and lipid profile) as a predictive factor for DEB risk in children and adolescents with T1D to suggest an early prevention plan for children and adolescents considered “at—risk”, as previous findings indicated that participating in a nutrition intervention to improve diet quality had no adverse effect on DEBs in this population [44]. The present study represents an innovative approach for the examination of nutritional habits and risk factors for developing DEBs in the T1D adolescent population. However, some limitations should be noted, including the sample size, which is considered modest, as LCA identified only 34 participants as belonging to the “at—risk” class. Furthermore, the use of a three-day self-reported food diary could represent a limitation in the assessment of daily dietary intake.

Among the strengths, the multi-center design allowed for the collection of participants’ data from different Italian regions. The use of the same analytical laboratory methods and food consumption measures, as well as the same macronutrient calculations, provided consistency. Furthermore, parents were trained before the study to maximize the reliability and limit potential bias in assessing dietary intake.

## 5. Conclusions

Our findings confirm that adolescence represents a delicate stage for T1D youth. LCA results highlighted a risk profile for DEBs characterized by vulnerabilities around both glycometabolic control and dietary intake. This study endorsed the importance of adopting a preventive approach in the daily clinical activity that may involve nutritional habits and diabetes management, stressing the fundamental role of specific diabetes education for youth with T1D and their families. Further studies are required to better investigate the association between dietary intake and DEBs in this population in order to shed light on the role of dietary patterns in predicting the development of eating disorders in youth with T1D.

## Figures and Tables

**Figure 1 nutrients-15-01721-f001:**
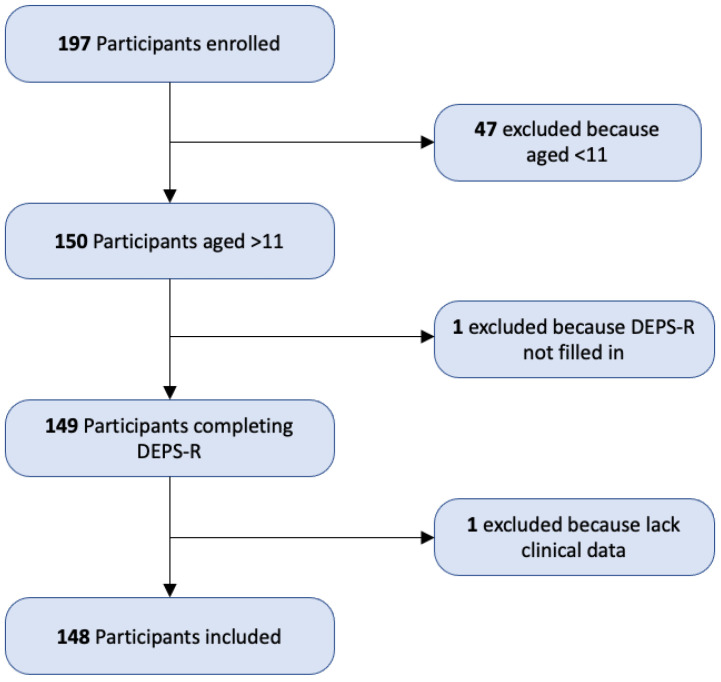
A flowchart of participants’ enrollment and inclusion.

**Figure 2 nutrients-15-01721-f002:**
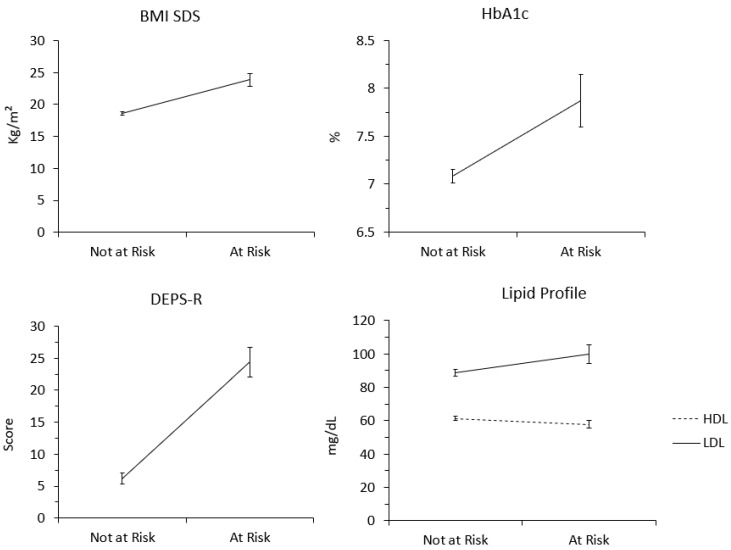
A latent class analyses graphical representation of class differences (divided for units of measurements).

**Figure 3 nutrients-15-01721-f003:**
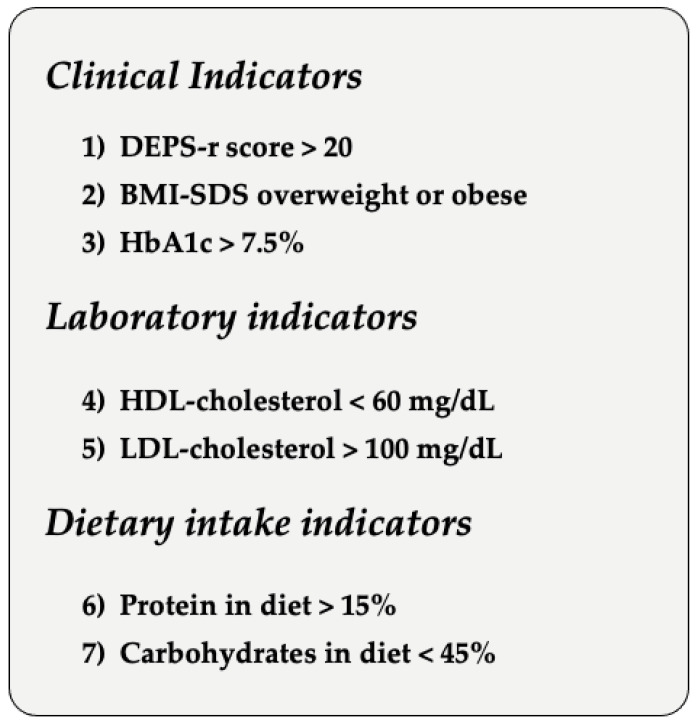
Characteristics of the “At—risk” profile for DEBs.

**Table 1 nutrients-15-01721-t001:** Clinical characteristics of the participants.

Children’s Variables	% (*n*)	Mean (SD)	Range
Ethnicity Italian African-American	98.0 (145) 2.0 (3)		
Age		12.1 (3.34)	3.3–17.8
Gender Female Male	48.0 (71) 52.0 (77)		
Diabetes Duration (years)		7.2 (3.4)	1.0–16.3
Insulin Therapy Modality MDI CSII	54.7 (81) 45.3 (67)		
Weekly Hours of Physical Exercise 0 1 2 3 More than 3	15.0 (22) 4.1 (6) 23.1 (34) 17.7 (26) 40.1 (60)		
Carbohydrate Counting Yes No	46.6 (69) 53.4 (79)		
Daily Insulin Dose		37.8 (19.2)	4.0–98.0
CGM data over 15 days Mean % Time < 54		0.6 (0.9)	0–7
Mean % Time 54–70		2.8 (2.9)	0–16
Mean % Time 70–180		57.9 (17.1)	15–96
Mean % Time 180–250		38.9 (18.3)	2–85
Mean % Time > 250		14.4 (12.3)	0–51

**Table 2 nutrients-15-01721-t002:** Nutritional habits.

Macronutrients	Mean % (SD)	Range %
Proteins	16.78 (3.29)	7.42–26.83
Saturated Fatty Acids	9.25 (2.53)	3.00–19.98
Monounsaturated Fatty Acids	16.44 (4.66)	4.52–31.81
Polyunsaturated Fatty Acids	9.91 (5.06)	1.82–31.45
Total Carbohydrates	45.05 (5.62)	29.64–56.87
Sugars	10.97 (3.83)	1.88–23.00

**Table 3 nutrients-15-01721-t003:** Demographics of the parents.

Parent’s Variables	% (*n*)	Mean (SD)	Range
Ethnicity Italian African-American	98.0 (145) 2.0 (3)		
Age Mothers Fathers		44.5 (6.2) 47.8 (6.3)	27.9–64.1 34.5–66.6
Parent Mothers Fathers	50.8 (148) 49.2 (143)		
Barratt Education Score Mothers Fathers		13.9 (4.9) 12.6 (5.3)	6–21 6–21
Barratt Occupation Score Mothers Fathers		21.3 (13.6) 25.2 (12.0)	5–45 5–45
Barratt Family SES Score		36.3 (13.6)	11–66
Family Annual Income <EUR 29,300 >EUR 29,300	37.8 (56) 62.2 (92)		
Type 1 Diabetes Familiarity Yes No	6.8 (10) 93.2 (138)		
Type 2 Diabetes Familiarity Yes No	11.5 (17) 88.5 (131)		

**Table 4 nutrients-15-01721-t004:** Latent class analysis.

LCA Models	Classes *n* (%)		SSA-BIC	Adjusted LMR-LRT	Entropy
	Class 1	Class 2			
One-Class solution	148 (100)		12,746.036	-	-
Two-Class solution	114 (77)	34 (23)	12,645.413	124.050	0.856
Variables		Class 1 Mean (*SD*)			Class 2 Mean (*SD*)
BMI-SDS		18.579 (0.242)			23.858 (0.950)
HbA1c		7.083 (0.071)			7.867 (0.274)
HDL		61.280 (1.221)			57.845 (2.378)
LDL		88.808 (1.952)			99.848 (5.382)
Protein		16.146 (0.269)			18.157 (0.606)
FSA		9.114 (0.200)			8.898 (0.608)
PUFA		10.914 (0.442)			8.462 (0.762)
MUFA		15.938 (0.371)			17.860 (1.103)
Carbohydrates		45.293 (0.464)			43.993 (1.457)
Sugar		10.748 (0.334)			10.605 (0.813)
DEPS-R Score		6.176 (0.836)			24.424 (2.325)

Notes. SSA BIC = Sample Size Adjusted Bayesian Information Criterium; Adj. LMR-LRT = Adjusted Lo–Mendell–Rubin Likelihood Ratio Test.

**Table 5 nutrients-15-01721-t005:** Differences in clinical variables between classes.

	Mean Values ‘Not At—Risk’ Class and ‘At—Risk’ Class	Sum of Squares	*df*	Mean Square	F	*p*
Age (years)	11.37–14.65	281.735	1	281.735	30.222	0.000
Diabetes Duration	6.74–8.70	99.831	1	99.831	9.283	0.003
Daily Insulin Doses	34.82–47.57	4255.898	1	4255.898	12.405	0.001
Number of Daily Insulin Injections	4.41–3.89	6.686	1	6.686	1.457	0.230
Number of Omitted Insulin Doses	0.13–0.22	0.186	1	0.186	0.797	0.373
Severe Hypoglycemic Episodes	0.03–0.12	0.218	1	0.218	2.207	0.140
DKA Episodes with Hospitalization	0.02–0.03	0.004	1	0.004	0.183	0.669
Parental SES	37.53–32.09	775.735	1	775.735	4.266	0.041
CGM data over 15 days Mean % Time < 54						
% Time < 54 mg/dL	0.69–0.31	3.661	1	3.661	4.014	0.047
% Time 54–70 mg/dL	3.07–1.93	33.958	1	33.958	4.214	0.042
% Time 70–180 mg/dL	58.8–55.16	346.657	1	346.657	1.184	0.278
% Time 180–250 mg/dL	37.85–42.46	558.348	1	558.348	1.677	0.197
% Time > 250 mg/dL	13.27–18.11	613.233	1	613.233	4.106	0.045

**Table 6 nutrients-15-01721-t006:** Results of the chi-square analysis.

Variables	Pearson Chi-Square Value	Likelihood Ratio	Linear-by-Linear Association	N of Valid Cases	*df*	*p*
*Gender*	0.15	0.15	0.15	148	1	0.903
*Therapy Modality*	0.57	0.57	0.57	148	1	0.811
*Carbohydrates Counting*	2.28	2.31	2.26	148	1	0.131
*T1D Familiarity*	1.76	1.56	1.75	148	1	0.185
*T2D Familiarity*	0.003	0.003	0.003	148	1	0.954

Note. *p*-values refer to a two-sided asymptotic significance for Pearson’s chi-square.

## Data Availability

Data are available from the corresponding author upon reasonable request.

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
