# Peer review of "A Risk Profile for Disordered Eating Behaviors in Adolescents with Type 1 Diabetes: A Latent Class Analysis Study"

_nutrients, 2023, doi:10.3390/nu15071721_

Round 1

Reviewer 1 Report

Interesting topic. Well described methods. A major limitation is the food diary for only 3 days and self reporting by individuals, which is very likely to be inaccurate in this cohort. Also the reporting of exercise may not be accurate and is not mentioned.

Readers familiarity with LCA technique of analysis may be a limitation, although it is described in detail.

Author Response

  • Interesting topic. Well described methods. A major limitation is the food diary for only 3 days and self – reporting by individuals, which is very likely to be inaccurate in this cohort.

We thank you for this useful comment. We have updated the study limitations according to your suggestion (marked in red).

As regards methods for registering food habits we have chosen a 3 – days food diary, as widely suggested in literature in this field1,2. This was also the most feasible way to assess dietary intake in this population, although we are aware a body of research suggests new tools that are more suitable for adolescents, such as Image – based assessment instruments. On the other hand, a recent study discussed the equal value of Image – based methods and food diaries in general as valid tools for assessing dietary intake3.

1 Handu D, Piotrowski M. Nutrition Interventions in Pediatric Patients with Type 1 Diabetes: An Evidence Analysis Center Scoping Review. J Acad Nutr Diet. 2022 Feb;122(2):424-431. doi: 10.1016/j.jand.2021.02.020. Epub 2021 Apr 14. PMID: 33865801.

2 Granado-Casas M, Solà I, Hernández M, Rojo-López MI, Julve J, Mauricio D. Effectiveness of medical nutrition therapy in adolescents with type 1 diabetes: a systematic review. Nutr Diabetes. 2022 Apr 22;12(1):24. doi: 10.1038/s41387-022-00201-7. PMID: 35459205; PMCID: PMC9033775.

3 Heikkilä L, Vanhala M, Korpelainen R, Tossavainen P. Agreement between an Image-Based Dietary Assessment Method and a Written Food Diary among Adolescents with Type 1 Diabetes. Nutrients. 2021 Apr 16;13(4):1319. doi: 10.3390/nu13041319. PMID: 33923638; PMCID: PMC8072648.

  • Also the reporting of exercise may not be accurate and is not mentioned.

Thank you for your comment.

We have now clarified the physical activity assessment method in the Methods section, according to your suggestion (marked in red).

This parameter was not considered as predictor for Latent Class Analyses (LCA) in this study, because the main focus was nutrition, food habits, nutritional status and glycemic control. As for many other variables in this study, physical exercise was included in the data collection as a descriptive variable to better understand the nature of our sample. We also included it in the ANOVA analyses to assess differences between the two latent classes of individuals and discussed the results in paragraph 3.3.

Reviewer 2 Report

The article by Giada Boccolini, et. al. entitled “Risk profile for eating disordered behaviors in adolescents with type 1 diabetes: a latent class analysis study” investigates the relationship between dietary intake, lipid profile, BMI-SDS and glycometabolic control, and disordered eating behaviors (DEBs) in adolescents (aged 11 to 18 years) with type 1 diabetes (T1D) to determine the risk profile for this group population. The authors used latent class analysis (LCA) to identify risk profiles, which allowed them to distinguish between two different classes of patients who were defined as "at risk" and "not at risk" for DEB. They indicate a decrease in BMI-SDS, HbA1c, LDL, proteins, monounsaturated fats in addition to an increase in HDL, saturated fats, polyunsaturated fats, carbohydrates, and sugar in adolescents in the "not at risk" group. Participants at risk have higher mean BMI-SDS, HbA1c, LDL, protein, monounsaturated fat, and DEPS-R, while they have lower averages for HLD, saturated fat, polyunsaturated fat, carbohydrates, and sugar. The authors propose an advanced approach to the study of dietary habits and risk factors for the development of disordered eating behaviors in adolescents with type 1 diabetes. Overall, this paper is very comprehensive, interesting, well-written, and contributes to increasing knowledge about the role of dietary patterns in predicting the development of eating disorders in youth with type 1 diabetes.

Author Response

Thank you very much. It was nice to read such a positive and responsive feedback by you.

So rewarding!